# Statistical Methods for Rapid Quantification of Proteins, Lipids, and Carbohydrates in Nordic Microalgal Species Using ATR–FTIR Spectroscopy

**DOI:** 10.3390/molecules24183237

**Published:** 2019-09-05

**Authors:** Lorenza Ferro, Zivan Gojkovic, András Gorzsás, Christiane Funk

**Affiliations:** Department of Chemistry, Umeå University, 901 87 Umeå, Sweden

**Keywords:** microalgae, FTIR spectroscopy, statistical methods, carbohydrates, lipids, proteins, nitrogen starvation

## Abstract

Attenuated total reflection–Fourier transform infrared (ATR–FTIR) spectroscopy is a simple, cheap, and fast method to collect chemical compositional information from microalgae. However, (semi)quantitative evaluation of the collected data can be daunting. In this work, ATR–FTIR spectroscopy was used to monitor changes of protein, lipid, and carbohydrate content in seven green microalgae grown under nitrogen starvation. Three statistical methods—univariate linear regression analysis (ULRA), orthogonal partial least squares (OPLS), and multivariate curve resolution-alternating least squares (MCR–ALS)—were compared in their ability to model and predict the concentration of these compounds in the biomass. OPLS was found superior, since it i) included all three compounds simultaneously; ii) explained variations in the data very well; iii) had excellent prediction accuracy for proteins and lipids, and acceptable for carbohydrates; and iv) was able to discriminate samples based on cultivation stage and type of storage compounds accumulated in the cells. ULRA models worked well for the determination of proteins and lipids, but carbohydrates could only be estimated if already determined protein contents were used for scaling. Results obtained by MCR–ALS were similar to ULRA, however, this method is considerably easier to perform and interpret than the more abstract statistical/chemometric methods. FTIR-spectroscopy-based models allow high-throughput, cost-effective, and rapid estimation of biomass composition of green microalgae.

## 1. Introduction

Microalgae are photosynthetic microorganisms able to convert water and carbon dioxide into valuable organic molecules by means of sunlight. Thanks to fast growth rates (doubling times in the order of hours) as well as minimal water and nutrients requirements, microalgal cultivation has high industrial and commercial potential for the sustainable production of biomass-derived fuels and chemicals, combined with the possibility of wastewater remediation and CO_2_ mitigation [1]. A wide range of algae-based products are already used in various sectors, including bioenergy, food and feed, green chemicals, and even cosmetics and therapeutics [2]. Three major classes of compounds are found in microalgae, which together contribute to more than 50% of their total biomass: amino acids (25–70%), carbohydrates (8–65%), and fatty acids/lipids (0–45%). Amino acids and proteins can be supplied as high nutritional value ingredients in the human diet and animal feed [3,4,5], but also used as organic biofertilizer to sustain crop productivity and preserve soil fertility [6,7]. Carbohydrates, including starch and polysaccharides, can be transformed into fermentable sugars for bioethanol production [8], used as emulsion stabilizer and bio-coagulant or as precursors for synthetic rubber and bioplastic generation [9,10]. Fatty acids and lipids can be converted into biodiesel through transesterification of triacylglycerides (TAGs) [11], but also constitute healthy food and feed supplements (e.g., omega-3 fatty acids) [3,4,5,12].

Due to the high metabolic flexibility of microalgal cells, the concentration of the major classes of compounds can drastically change in the produced biomass, depending on the culturing conditions [13,14]. For example, nutrient deprivation, especially nitrogen starvation, has been shown to stress microalgal cultures and induce an intra-cellular accumulation of energy-rich molecules as a result of the reallocation of photosynthetically fixed carbon under limiting nutrient conditions [15,16]. Microalgae can dramatically increase their lipid and/or carbohydrate content in the matter of days or even hours via carbon partitioning, which appears to be species-specific [17,18]. To follow these rapid changes, a fast, simple, and cost-effective method is needed to enable effective monitoring of microalgal cultures, rapidly selecting highly productive strains, and understanding their metabolic behavior under different growing parameters and nutrients deficiency conditions.

Fourier transform infrared (FTIR) spectroscopy has been shown to be useful for the analysis of chemical compounds in biological samples, including microbial biomass, for intra- and extracellular metabolites [19,20,21] and microalgal biomass [22,23,24]. FTIR spectra give rise to characteristic bands, which reflect the biochemical composition of the sample. Qualitative and quantitative differences in the molecular composition of the sample are deduced by the position (traditionally expressed in wavenumbers (cm^−1^) instead of wavelength) and intensity (the ‘height of’ or ‘area under’) of these characteristic bands.

Each microalgal strain has been shown to produce a unique FTIR spectrum, with specific peaks occurring at defined wavelengths, which is dependent on the environmental conditions [23,25]. So far FTIR spectroscopy has been generally used for semi-quantitative screening only, i.e., the determination of the relative abundances of molecular compounds in microalgal biomasses [26,27,28]. Only a few studies have explored the possibility of absolute quantification of the microalgal components by FTIR spectroscopy [29,30]. These studies were limited to one (or maximum a few) model species, all cultivated under constant growth conditions, and the calculated concentrations were mainly derived from linear regression analysis after calibration against external standards as reference substances. These factors, coupled to the complexity of the spectra, including overlapping of bands, as well as the dependency on sample cell size and external standards might generate misleading and inaccurate quantifications under large scale ‘real’, varying growth conditions, where the chemical biomass composition changes both qualitatively and quantitatively (i.e., microalgae will produce different chemical compounds, not only different amounts of the same compounds).

In this study, ATR–FTIR (attenuated total reflection–FTIR) spectroscopy was used to quantitatively assess species-specific changes of protein, lipid and carbohydrate content in the biomass of six natural Nordic microalgal strains [31] and one culture collection (control) strain during progressive nitrogen starvation. During ATR–FTIR spectroscopic measurements, infrared radiation reflected by a crystal with high refractive index is partially absorbed by the sample and finally collected by a detector [32,33]. In the current investigation, freeze-dried algal biomass was directly applied on the diamond crystal to avoid time-consuming sample pre-processing plaguing other FT–IR spectroscopic techniques, which often even cause non-uniform samples. Minimal to no sample preparation in ATR–FTIR spectroscopy allows high throughput (up to 30 samples/hour) and simultaneously gives a plethora of information on the biochemical composition of each sample. ATR–FTIR spectroscopy therefore is a superior alternative to the traditional analytical methods, which are generally more laborious, time-consuming, error-prone, and often require expensive and harmful reagents.

Three different statistical methods—univariate linear regression analysis (ULRA), orthogonal partial least squares (OPLS), and multivariate curve resolution alternating least squares (MCR–ALS)—were compared as mathematical tools to predict the amount of proteins, lipids, and carbohydrates in the algal biomass from the collected spectra; the robustness and applicability of the models were evaluated by comparing the predicted values with values received after classical chemical extractions. ULRA and OPLS models were built based only on empirical and spectral data from microalgal biomass measurements, whereas MCR–ALS, a statistical approach here applied for the first time to resolve microalgal IR spectra, was modeled using reference substances (i.e., albumin, algal lipid extract, and cellulose, to represent proteins, lipids, and carbohydrates, respectively) as initial pure spectral components. The proposed FTIR spectroscopy-based models can find application in cost-effective and rapid estimation of biomass composition of any green microalgal species.

## 2. Results and Discussion

### 2.1. Algal Biomass Composition Based on Classical Extractions and ATR–FTIR Spectral Analysis

The biomass of six natural Nordic microalgal strains and a culture collection strain exposed to N-starvation was analyzed after classical extraction and compared to data received by ATR–FTIR spectroscopy. While the classical extraction of proteins was relatively fast and easy to perform (few steps, short incubation times) and gave accurate results, methods to extract lipids and carbohydrates were more laborious and involved error-prone critical steps (carbohydrate hydrolysates for example are hard to dissolve in the phenol-sulfuric acid solution prior to spectrophotometric analysis). The extraction efficiency varied from one species to another and between culture stages of the same species, making estimations of the concentrations difficult and leading to further experimental errors. Indeed, chemical extraction of compounds from microalgae can be difficult due to one to multiple thick cell walls surrounding the algae, which are particularly hard to break for some species. Additionally, compounds (e.g., sporopollenin) have been identified in the cell wall of several green microalgae, which act as a protective barrier against chemical and biological degradation due to their recalcitrant nature [34,35]. Pre-treatment of the biomass is therefore required but not always efficient and suboptimal extraction is leading to underestimated quantities [36,37]. Furthermore, extraction and purification steps required in the traditional chemical methods lead to further losses and increase the uncertainty of the measurements due to potential errors at each step.

With regard to microalgal biomass composition, as expected the protein content decreased along the N starvation period in all seven algal species, while the content of lipids or carbohydrates increased. The partitioning of carbon into these two storage compounds was, however, species dependent; while *C. vulgaris* 13-1 preferentially accumulated lipids, *Scenedesmus* sp. B2-2 mainly accumulated carbohydrates, and *Desmodesmus* sp. RUC-2 accumulated both compounds. Details on daily nitrogen removal from the growth medium and measured concentrations of proteins, lipids, and carbohydrates in the algal biomass (from day 0 of nitrogen starvation) are reported in the Appendix A (Table A1 and Table A2, respectively). The concentration of the compounds (as% DW) in the biomass ranged from 9–48% for proteins, 0–13% for lipids, and 13–55% for carbohydrates.

ATR–FTIR spectral data were rapidly collected from freeze-dried algal biomass samples (and reference compounds) and processed. No sample preparation was needed, this approach therefore is highly time-efficient and easy. Despite the intrinsic complexity of the FTIR spectra derived from biological samples [38], the characteristic bands for each compound of interest (at ~1745 cm^−1^ for lipids, ~1650 cm^−1^ for proteins, and ~1010 cm^−1^ for carbohydrates) identified in the reference spectra could easily be detected in all spectra originating from biomass of the algal species (Figure 1). Compound amounts differing between spectroscopic analysis and biochemical extraction were considered as outliers and excluded from the statistical analyses: the amount of lipids and carbohydrates in *C. astroideum* RW10 at the end of the N starvation period (day 6, 8) appeared overestimated, likely due to a carryover of impurities after chemical extraction, while the concentration of carbohydrates in *C. vulgaris* 13-1 at day 0 of the N starvation period was probably underestimated using this method, likely due to inefficient cell wall hydrolyzation. The exclusion of these outliers (Table A2) considerably improved the fitting of data and predictive power of the models (i.e., higher R^2^ and Q^2^) for lipids and carbohydrates with all the statistical methods tested (data not shown). FTIR spectra of the excluded samples are given in the Appendix A (Figure A1). While spectra from the same culture collected in previous and consecutive sampling match, the empirical values deteriorate. This points to an experimental error in the empirical values rather than in the spectral recording.

### 2.2. Statistical Methods (ULRA, OPLS, and MCR–ALS) Can Facilitate the Prediction of Protein-, Lipid- and Carbohydrate-Content in Microalgae

Three different statistical approaches were used to build models based on the processed ATR–FTIR spectral data, derived from the biomass of the seven different green microalgal species at different stages of cultivation under nitrogen starvation. The intensities (determined as the area under the peak) of diagnostic bands for proteins, lipids, and carbohydrates were used for univariate linear regression analysis (ULRA), whereas the entire spectra in the fingerprint region (800–1800 cm^−1^) were considered for orthogonal partial least squares (OPLS) and multivariate curve resolution alternating least squares (MCR–ALS) analyses. Independent predictions were then obtained by testing the models with full cross validation (LOO-CV) and, for ULRA and OPLS, with an external dataset (samples from the microalga *Desmodesmus* sp. 2-6, not included when building the models). Predicted values of the three compounds of interest were plotted against their values derived from classical chemical extractions, and both model accuracy and predictive ability were evaluated.

#### 2.2.1. Method 1: ULRA, based on FTIR spectral band intensities

The ULRA model built using FTIR spectral band intensities gave good correlations (R^2^ > 0.8) for proteins and lipids (Figure 2a,b), with relatively small errors for both calibration (RMSEC = 3.2, 1.25) and internal (RMSECV = 3.4, 1.3) and external validations (RMSEP = 3.2, 0.7) (Table 1).

Carbohydrates, however, were more difficult to model: a rather low correlation between predicted and actual values (R^2^ = 0.65, Figure 2c) was obtained resulting in a high error (RMSEC = 4.4). This model therefore has poor predictive performance (RMSECV = 4.6), especially for new data (RMSEP = 9.3) (Table 1). The residual prediction deviation (RPD) was calculated to compare the robustness and validity between the models, based on their cross validation. Minimum RPD values required for possible quantitative predictions lay between 2–2.5; higher values indicate good (2.5–3) or excellent (>3) prediction accuracy [39]. As shown in Table 1, the ULRA models built for proteins and lipids had good (RPD = 2.7) and acceptable (RPD = 2.2) predictive abilities, respectively, whereas the accuracy of the model built for carbohydrates was unsatisfactory (RPD = 1.6). The observed difficulties to model carbohydrates are related both to X and Y variables; limitations in chemical extraction and quantification of carbohydrates result in higher errors of the Y variables. Errors in the X variable arise from the baseline correction and normalization of the ATR–FTIR spectra. ATR–FTIR spectral bands are proportionally more intensive at lower wavenumbers as compared to transmission FTIR spectra, because of the increased penetration depths of the IR radiation at lower wavenumbers (higher wavelength). Carbohydrates, absorbing in the lowest wavenumber region of all three major classes of compounds (960–1130 cm^−1^) will therefore give rise to relatively stronger bands compared to proteins and especially lipids, which absorb at shorter wavelengths (higher wavenumbers, at 1660 and 1740 cm^−1^, respectively). Furthermore, the region in the FTIR spectrum corresponding to carbohydrates is highly complex, consisting of a large number of overlapping bands, some of them having contributions from the functional groups of other molecules (e.g., stretching of PO_2_ and Si-O groups) [30]. These factors, together with potential errors in baseline correction and normalization, might lead to inaccurate estimations of the amount of carbohydrates. However, calculating the carbohydrate/protein ratio for both FTIR spectral band intensities improved the ULRA model considerably compared to the concentrations determined after extractions; the explained variance increased to R^2^ = 0.84 (Figure 2d) and the error decreased even for the independent predictions (RMSEC, RMSECV, RMSEP < 0.5 and RPD = 2.5, Table 1). Using the accuracy of the ULRA model to predict the amount of proteins therefore allows to indirectly estimate the carbohydrate content.

#### 2.2.2. Method 2: OPLS Based on the Fingerprint Region of FTIR Spectra

FTIR spectra derived from biological material (e.g., whole cells) typically present superimposed spectra of individual chemical components, which can be difficult to deconvolute. Multivariate data analysis is a powerful tool for the interpretation of these complex spectral data and allows prediction of the chemical composition in the sample by reducing its dimensionality [33]. In principal component analysis (PCA), the maximal variation of the data is identified along principal components (latent variables), which can be used to detect spectral (and thus chemical) differences between samples. Analysis of PCA loadings identifies the specific spectral regions responsible for these differences [40,41]. In order to find a relation between principal components and independent variables (i.e., analytical data from e.g., classical chemical extractions), partial least squares (PLS) regression can be used. PLS builds models for both X (spectral) and Y (independent) variables and then maximizes their correlation. OPLS is a variant of PLS analysis, in which the orthogonal variance in X (i.e., the variation that is not correlated to Y) is also considered, thus improving interpretability and predictions of the model [42].

Our OPLS model was able to discriminate the algal samples based on their ATR–FTIR spectra, and thus their biochemical composition, explaining more than 98% of the variance in X and 86% of the variance in Y (Table 1). The model was built using five significant principal components: two predictive components, two orthogonal components in X, and one orthogonal component in Y. The first predictive component, explaining most of the variation in the samples (R^2^X = 0.78, R^2^Y = 0.66) was positively correlated (>75%) to the spectral region assigned to proteins (1500–1700 cm^−1^) and negatively correlated (>75%) to the region assigned to carbohydrates (980–1100 cm^−1^); the second predictive component was positively correlated (>75%) to the band assigned to lipids (1700–1780 cm^−1^) and negatively (>50%) to specific bands in the carbohydrate region (Appendix A, Figure A2). Thus, the first component can differentiate the biomass of stressed and non-stressed microalgae, as non-stressed cells prevalently contain proteins, while storage compounds (such as carbohydrates) are progressively accumulated during nitrogen starvation. Furthermore, the second component was able to predict which one of the two storage compounds (lipids or carbohydrates) was preferentially produced by different microalgal species during the nitrogen stress and points out a change in the composition (nature) of carbohydrates, not only their relative amounts. The prediction accuracy of the total variation of X and Y (based on 5-fold CV) was high (Q^2^ > 0.80), demonstrating the robustness of the model, which has the advantage to include and consider all three *y*-variables (i.e., all three classes of compounds) simultaneously.

When modeled separately, excellent correlation coefficients (R^2^Y ≥ 0.90) were found for single *y*-variables of both proteins and lipids (Figure 3a,b), and a good correlation coefficient (R^2^Y = 0.77) for carbohydrates (Figure 3c). Errors occurring from calibration and internal validation were smaller compared to the corresponding ULRA models: the error occurring from external validation was slightly higher for lipids (RMSEP = 1.1), but far lower for proteins (RMSEP = 1.5) and carbohydrates (RMSEP = 4.1) (Table 1). The accuracy of the model assessed by RPD statistic revealed a very good predictive ability for proteins (RPD = 3) and lipids (RPD = 2.9), but was also acceptable for carbohydrates (RPD = 2) without referencing it to protein concentrations first. Thus, quantitative prediction of carbohydrates is also possible using the OPLS method.

#### 2.2.3. Method 3: MCR–ALS Based on the Fingerprint Region of FTIR Spectra

Three pure component spectra from reference substances (proteins: BSA, lipids: microalgal extract, carbohydrates: MCC, Figure 1b) were manually supplied as initial estimates to calculate the corresponding relative concentration profiles in our algal biomass samples using MCR–ALS. In this statistical approach, the original complex spectra are decomposed into a set of ‘pure’ components, resulting in both spectral profiles and concentration estimates for these components [43]. It has to be noted, however, that ‘pure’ components do not necessarily mean pure chemical compounds, as the resolving power of MCR–ALS greatly depends on the dataset. Indeed, in our case, MCR–ALS was unable to correctly unmix the complex spectra from the original data set via singular value decomposition: the contribution of each of the pure identified components resulted in overlapping peaks especially in the carbohydrate and protein regions (data not shown). Using the spectra of reference substances as initial estimates helped the MCR–ALS algorithm to reach an endpoint with purer profiles.

We applied lack of fit (lof) and explained variance (R^2^) to evaluate the quality of the model and received a very good fit (R^2^ > 99%), with relatively low uncertainty (lof PCA = 0.8% lof exp. = 5.9%) using the three major classes of compounds as components (Table 1). The model’s prediction ability on the concentration of proteins, lipids, and carbohydrates in the algal samples was evaluated by fitting the calculated concentration profiles for each of the three pure components with the values received by extractive methods (Figure 4). We received results similar to ULRA: the best model was obtained to quantify the protein amount, with good correlation (R^2^ = 0.85) and low error values for calibration (RMSEC = 3.5) and cross-validation (RMSECV = 3.7); acceptable correlation (R^2^ = 0.77) and error values (RMSEC = 1.3, RMSECV = 1.4) were obtained for lipid quantification, whereas carbohydrates could only be estimated approximately (R = 0.63, RMSEC = 4.51, RMSECV = 4.73). Similar to the ULRA model, high complexity and proportionally higher intensity of the carbohydrate region in ATR–FTIR spectra hinder correct quantification of carbohydrates.

It is important to keep in mind that MCR–ALS *per se* is not an entirely quantitative method due to potential rotational ambiguities. Nevertheless, it can still offer a bilinear description of the data and can provide meaningful models to semi-quantitatively estimate chemical components in the samples. Most importantly, MCR–ALS has the advantage of being performed and interpreted without much effort; it provides spectral and concentration profiles, which are easy to validate. Furthermore, it can be performed by open-source, user-friendly graphical interfaces ([44], https://www.umu.se/en/research/infrastructure/visp/).

## 3. Materials and Methods

### 3.1. Algal Cultivation and Sampling Preparation

Samples of algal biomass used in this work were obtained from six different natural green microalgae previously isolated in Sweden (*Chlorella vulgaris* 13-1, *Coelastrella* sp. 3-4, *Coelastrum astroideum* RW10, *Desmodesmus* sp. RUC-2, *Scenedesmus* sp. B2-2, and *Desmodesmus* sp. 2-6) and one culture collection strain (*Scenedesmus obliquus* UTEX 417) [31]. The microalgae were cultivated in duplicate batch experiments (R1, R2) in flat panel photobioreactors (width × height × depth = 30 × 30 × 1.5 cm) [45] illuminated from one side by white LED panel with adjustable light intensity of 45–650 µmol photons/m^2^/s under photoautotrophic conditions for 12 days in BBM [46] as cultivation medium containing an initial nitrogen concentration of 5 mM. The algal cultures were homogenously mixed by bubbling a 3% *v*/*v* CO_2_/air mixture through a silicon tube with small holes placed horizontally at the bottom of the bioreactor. N consumption during algal cultivation was regularly monitored by chemical reduction (Griess reaction followed by vanadium chloride oxidation in concentrated HCl) and spectrophotometric analysis [47]. After complete N depletion from the medium (day 0) samples were collected every second day to evaluate changes in protein, lipid and carbohydrate content under N starvation. Biomass was separated from the medium by centrifugation and freeze-dried overnight for downstream analyses (chemical extractions and FTIR spectroscopy).

### 3.2. Chemical Extraction of Proteins, Lipids, and Carbohydrates

The protein content was determined by total protein precipitation [48] followed by colorimetric assay and spectrophotometric quantification. Approx. 2 mg of freeze-dried algal biomass were dissolved in 200 µL of a 24% trichloroacetic acid (TCA) solution (*w*/*v*), vortexed and incubated at 95 °C for 15 min. The colorimetric assay was performed using the DC Protein Assay kit (BIO-RAD) following manufacturer’s instructions and the protein absorbance was read at 750 nm in a spectrophotometer (Varian Cary 50 Bio, Agilent Technologies). Proteins were quantified based on a calibration curve built with different concentrations of bovine serum albumin (BSA) as protein standard.

The lipid content was determined as described by [15]. Briefly, approx. 5–15 mg of freeze-dried biomass was dissolved in a 4:5 (*v*/*v*) chloroform: methanol solution with glass beads and treated in a bead beater (Bullet Blender homogenizer, Next Advance, USA) to achieve cell disruption. The extracted lipids were resuspended in 7:1 (*v*/*v*) hexane: diethyl ether, purified through pre-equilibrated silica columns (Sep-Pak Silica Vac cartridges, Waters), methylated in a methanol/H_2_SO_4_ solution for 3 h at 70 °C and quantified in a gas chromatograph (TRACE 1310 GC, Thermo Scientific) using a 30 m column (FAMEWAX, Restek Corporation) and nitrogen as carrier gas.

The carbohydrate content was determined by hydrolysis followed by phenol-sulfuric acid extraction and a colorimetric essay [49]. Hydrolysis of approx. 2 mg of freeze-dried biomass was performed in HCl at 90 °C for 3 h. After neutralization with NaOH, solutions of 5% phenol (*v*/*w*) and 0.45 H_2_O: 2.5 H_2_SO_4_ (*v*/*v*) were added to 0.05 mL of samples, following incubation at 35 °C for 30 min prior spectroscopic measurements at 483 nm. Carbohydrates were quantified based on a calibration curve built with different concentrations of glucose as sugar standard.

Concentrations of proteins, lipids, and carbohydrates were calculated and expressed as percentage of biomass dry weight (% DW).

### 3.3. ATR–FTIR Spectroscopy and Spectral Data Processing

The microalgal biomass was analyzed by attenuated total reflectance–Fourier transform infrared (ATR–FTIR) spectroscopy using a Bruker Vertex 80v spectrometer (Bruker Optik GmbH, Ettlingen, Germany) under vacuum conditions, equipped with a Bruker PLATINUM ATR accessory and diamond internal reflection element and a deuterated-triglycine sulfate (DTGS) detector. A few milligrams of freeze-dried biomass were directly applied and pressed on the crystal plate and raw spectra were recorded in the range of 400–4000 cm^−1^ (128 scans per sample, 4 cm^−1^ spectral resolution) using OPUS (version 6.5). Additionally, infrared spectra from pure BSA (bovine serum albumin), microcrystalline cellulose (MCC) and total lipids extracted from the microalga *S. obliquus* UTEX 417 (methanol extract) were collected and used as reference spectra for proteins, carbohydrates, and lipids, respectively. The diamond crystal was carefully cleaned with ethanol before each measurement to avoid carryover of biomass from previous samples. Interpretation of the spectral data was based on band assignments described previously [50]: peaks in the range 980–1200 cm^−1^ were assigned to carbohydrates, deriving from ring vibrations of carbohydrates and asymmetric -C-O-C- stretch in polysaccharides; peaks at ca. 1660 cm^−1^ and 1540 cm^−1^ (amide I and amide II vibrations from C=O stretching vibrations and N–H bending vibrations of peptide bonds, respectively) were used to estimate protein content; and the peak between 1710–1760 cm^−1^ was assigned to lipids and fatty acids (deriving from non-peptide -C=O stretches).

Raw infrared spectra collected from 58 samples were imported in MATLAB and processed using the free, open source MATLAB-based script provided by the Vibrational Spectroscopy Core Facility at Umeå University (www.umu.se/en/research/infrastructure/visp/downloads/). Spectra were baseline corrected by asymmetric least squares (AsLS λ = 1000000; AsLS *P* = 0.001), total area normalized and mildly smoothed (Savitzky-Golay filter with a first order polynomial and a frame of 5). Only the fingerprint region between 800–1800 cm^−1^ was considered for downstream analyses, being the most informative for algal biomass composition [25] and least sensitive to potential baseline correction and normalization errors. Infrared spectral band intensity analysis was performed by measuring the area under the absorption bands diagnostic for proteins (1580–1700 cm^−1^), lipids (1710–1765 cm^−1^), and carbohydrates (960–1130 cm^−1^), using the built-in function of the same MATLAB-based script used for processing the spectra.

### 3.4. Modeling and Statistical Analysis

Three different statistical methods were used to model the relationship between experimental data (measured concentration via chemical extraction) and spectral data and evaluate the predictive power of the FTIR spectroscopic analysis on protein, lipid, and carbohydrate content in microalgal biomass: univariate linear regression analysis (ULRA), orthogonal partial least squares (OPLS), and multivariate curve resolution alternating least squares (MCR–ALS). Data obtained from the microalgal species *Chlorella vulgaris* 13-1, *Coelastrella* sp. 3-4, *Coelastrum astroideum* RW10, *Desmodesmus* sp. RUC-2, *Scenedesmus* sp. B2-2, and *S. obliquus* UTEX 417 (*n* = 52) were used as calibration set to build the models. Data collected from the microalgal species *Desmodesmus* sp. 2-6 (*n* = 6) were used as external validation set in ULRA and OPLS methods to test the universality of the models.

#### 3.4.1. ULRA

Univariate linear regression models were inferred by using experimental data (% DW) as dependent variable and the infrared spectral band intensity (i.e., integrated area under the absorption bands) as independent variable. The statistical software R was used for data processing (model calibration and validation) [51]. Strong outliers clearly deviating from the regression model and likely resulting from poor chemical extraction (*n* = 2 for lipids, *n* = 6 for carbohydrates) were removed from the corresponding datasets. The predictive power of each model was evaluated using the leave-one-out cross validation (LOO-CV) method (internal validation). The following statistical parameters were calculated: intercept and slope coefficient; R^2^ (coefficient of determination), estimator of the goodness of fit; RMSEC (root mean square error of calibration), estimator of the predictive ability of the model based on the calibration dataset; Q^2^, i.e., the predictive R^2^, and RMSECV (root mean square error of cross validation), estimators of the predictive ability of the model based on LOO-CV; RPD (residual predictive deviation), a qualitative estimator of the model predictions calculated as the ratio of the dependent variable SD to RMSECV; RMSEP (root mean square error of prediction), estimator of the predictive ability of the model on the external dataset (external validation). RMSEC, RMSECV, and RMSEP were calculated as
(1)$$RMSE\left(C,\;P\right)=\sqrt{\frac{\sum_{i=1}^{n}{\left({\hat{y}}_{i}-y_{i}\right)}^{2}}{n}}RMSECV=\sqrt{\frac{\sum_{i=1}^{n}{\left({\hat{y}}_{i}-y_{i}\right)}^{2}}{n-1}}$$
where *ŷ_i_* represents predicted values (from the model) and *y_i_* represents values from chemical extractions.

#### 3.4.2. OPLS

A dataset including processed infrared spectra and estimated concentrations of proteins, lipids, and carbohydrates for the 52 algal biomass samples was imported to SIMCA-P (v. 15, Umetrics AB, Sweden). Sample names were set as primary ID; wavelengths (cm^−1^) and names of biochemical compounds (proteins, lipids, carbohydrates) were set as secondary ID; infrared spectral intensities (at 2 cm^−1^ intervals, following zero-filling) represented X-variables (independent) and were scaled to center (Ctr); estimated concentrations of biochemical compounds represented *y*-variables (dependent) and were scaled to unit variance (UV). A single OPLS model was created including all three *y*-variables. Outliers were excluded as in ULRA. The number of significant components was calculated by k-fold cross validation, with *k* = 5. The following statistical parameters were calculated: number of significant components (predictive, orthogonal in X and Y); R^2^X (cum) and R^2^Y (cum), the cumulative fraction of X and Y variation explained by the model; RMSEC (equivalent to RMSEE in SIMCA); Q^2^ (cum), the cumulative fraction of Y variation predicted by the model according to cross-validation, RMSECV, RMSEP.

#### 3.4.3. MCR–ALS

Concentration profiles of proteins, lipids and carbohydrates in the algal biomass samples were determined by MCR–ALS analysis using the free, open-source MATLAB script by the Vibrational Spectroscopy Core Facility at Umeå University (www.umu.se/en/research/infrastructure/visp/downloads/). Infrared spectra from the biomass samples were imported and processed as described above. The three reference spectra were imported and used as initial estimates and MCR–ALS modeling was performed with 50 iterations and 0.1 convergence limit (default values). The following statistical parameters were calculated at the end of the modelling: lack of fit for PCA and for the experimental variation (%) at optimum; R^2^ (%) at optimum. Lack of fit is defined as the difference among the input data and the data reproduced by MCR–ALS. This value is calculated according to the expression
(2)$$lack\;of\;fit\;\left(\%\right)=\sqrt{\frac{\sum_{i,j}^{n}e_{ij}{}^{2}}{\sum_{i,j}^{n}d_{ij}{}^{2}}}$$
where *d_ij_* represents an element of the input data matrix and *e_ij_* is the related residual obtained from the difference between the input element and the MCR–ALS reproduction. The two lack of fit values calculated are differing by the input data matrix D used: either the raw experimental data matrix or the PCA reproduced data matrix using the same number of components as in the MCR–ALS model [43]. Estimates of proteins, lipids, and carbohydrates obtained by MCR–ALS analysis of spectral data and their measured concentrations by chemical extractions were fitted and R^2^ calculated to measure the strength of the linear relationships.

## 4. Conclusions

ATR–FTIR spectroscopy is a simple, fast, and cost-efficient technique that allows to collect a broad range of information on the chemical composition of complex biological samples. The method is non-destructive and requires neither extraction nor external agents (labels, dyes, or markers). We applied this method to quantitatively monitor changes in the content of proteins, lipids, and carbohydrates of locally isolated green microalgal species at different stages of cultivation under progressive nitrogen starvation. Considerable time could be saved by being able to directly measure very small quantities of dried biomass without any pre-treatment/preparation, allowing high-throughput screening of algal cultures. We tested three alternative statistical methods (ULRA, OPLS, and MCR–ALS) with good statistical confidence to address accuracy and prediction ability. Generally, all models showed good correlations between spectral and empirical data, good prediction abilities for the concentrations of proteins and lipids, and acceptable approximations for the concentrations of carbohydrates. The prediction of carbohydrate concentration has room for improvement, especially via MCR–ALS modeling, e.g., by exploring a wider range of constraints. The OPLS method, however, was able to model all three components simultaneously with high prediction ability (even for carbohydrates) and was therefore the most robust approach. OPLS further provided insights in the biomass composition in relation to the algal cultivation stage and displayed the preferential storage compound accumulating in the algal cells. Potential application areas of this method are wide, including the screening of new algal strains and the evaluation of species-specific metabolic response to different environmental stresses for boosting the production of specific classes of compounds.

## Figures and Tables

**Figure 1 molecules-24-03237-f001:**
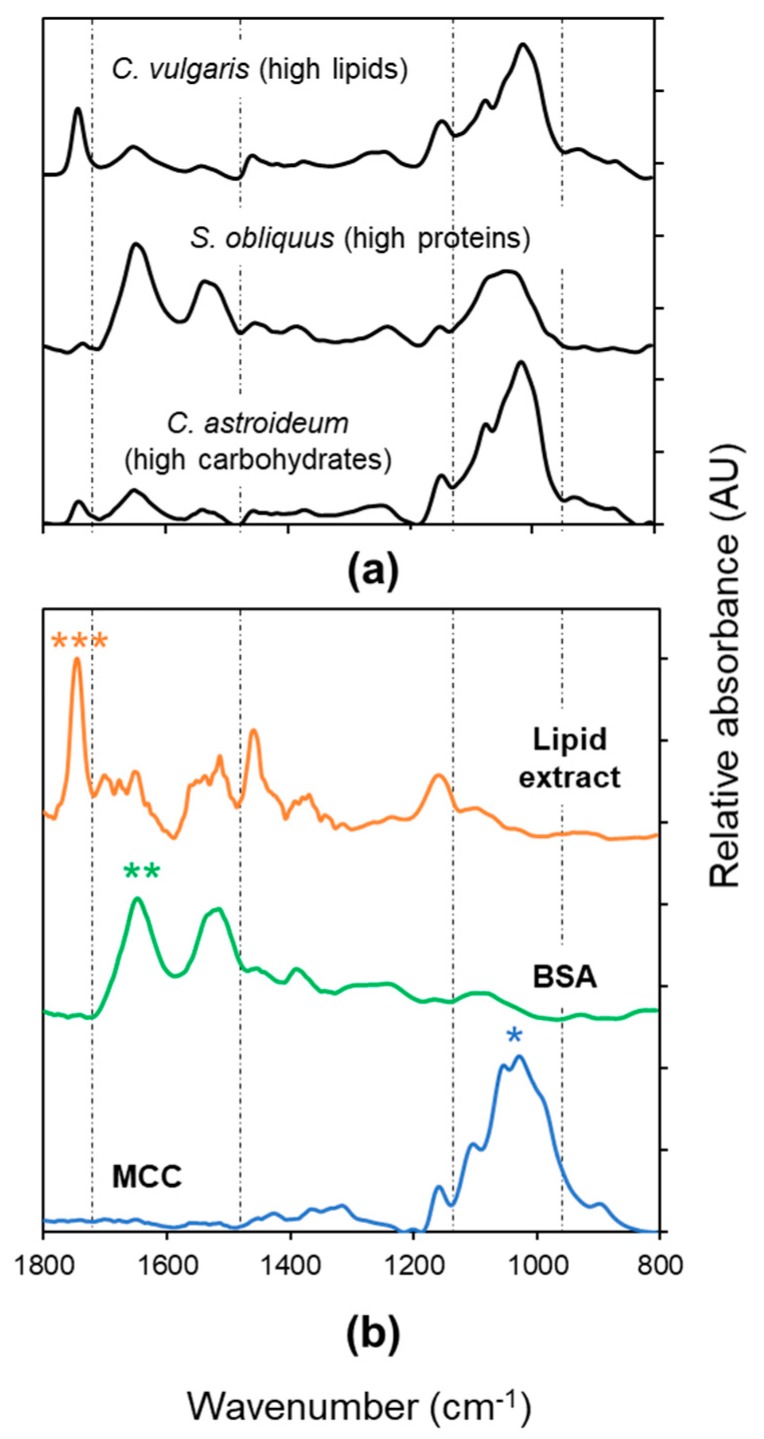
Examples of ATR–FTIR spectra (**a**) from the biomass of three microalgal species (*C. vulgaris* 13-1, *S. obliquus* UTEX 417, *C. astroideum* RW10) taken at harvest (day 6 or 8 after nitrogen starvation), and reference spectra (**b**) for lipids (extract from *S. obliquus* UTEX 417), proteins (bovine serum albumin, BSA) and carbohydrates (microcrystalline cellulose, MCC). Diagnostic bands are marked for carbohydrates (one star (*), 960–1130 cm^−1^); proteins (two stars (**), 1580–1700 cm^−1^) and lipids (three stars (***), 1710–1765 cm^−1^).

**Figure 2 molecules-24-03237-f002:**
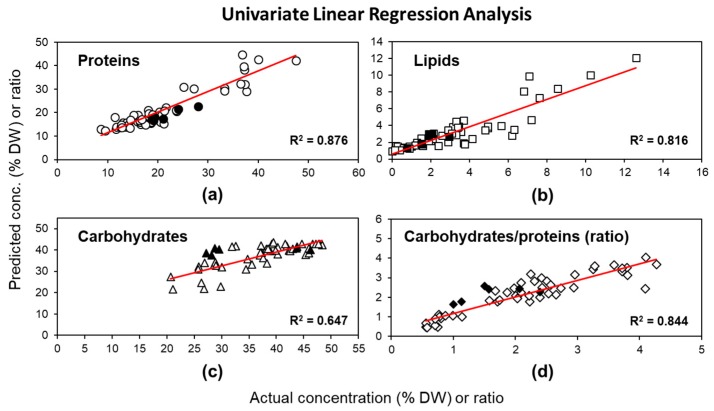
Correlation between determined (by classical extractions) and predicted concentrations (as percentage of dry weight, DW) of (**a**) proteins, (**b**) lipids, (**c**) carbohydrates, and (**d**) the ratio of carbohydrates to proteins, based on ULRA models of FTIR spectral band intensities. R^2^ represents the coefficient of determination of linear regression. White symbols indicate data points of the calibration set; black symbols indicate data points of the external validation set.

**Figure 3 molecules-24-03237-f003:**
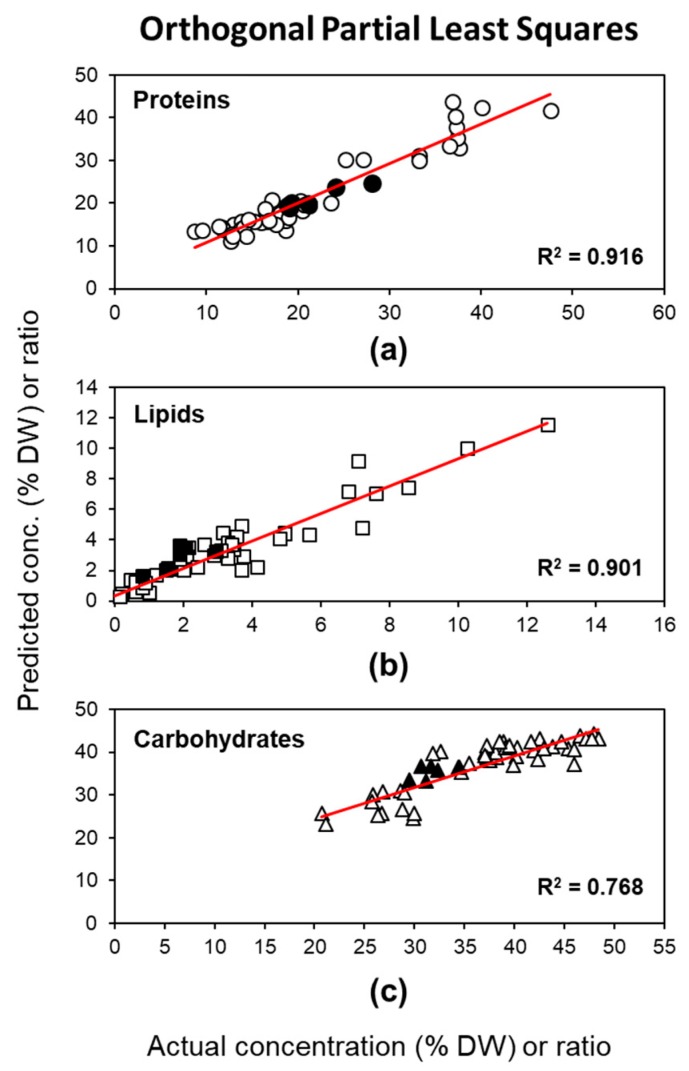
Correlation between determined (by classical extractions) and predicted concentrations (% DW) of (**a**) proteins, (**b**) lipids, and (**c**) carbohydrates based on the OPLS model of the fingerprint region of FTIR spectra. R^2^ represents the coefficient of determination of linear regression. White symbols indicate data points of the calibration set; black symbols indicate data points of the external validation set.

**Figure 4 molecules-24-03237-f004:**
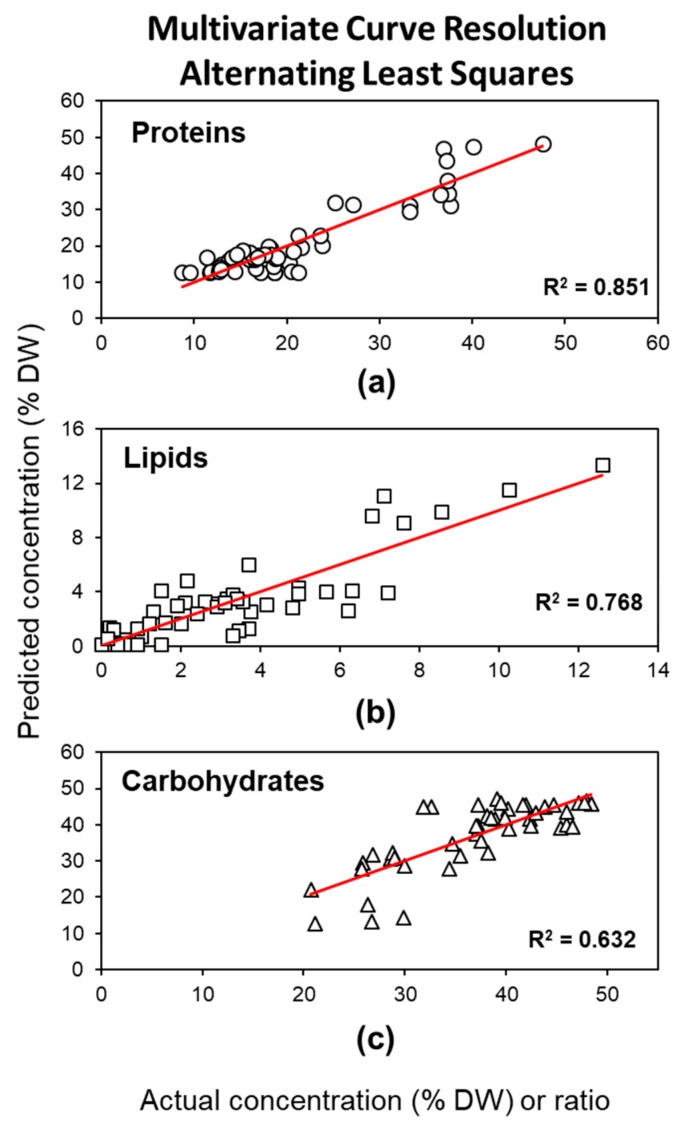
Correlation between determined (by classical extractions) and predicted concentrations (% DW) of (**a**) proteins, (**b**) lipids, and (**c**) the carbohydrate to protein ratio, based on MCR–ALS resolved concentration profiles for proteins, lipids, and carbohydrates. R^2^ represents the coefficient of determination of linear regression. White symbols indicate data points of the calibration set.

**Table 1 molecules-24-03237-t001:** Comparison of the models ULRA, OPLS, and MCR–ALS to predict the content of proteins, lipids, and carbohydrates in microalgal biomass. N: number of samples; R^2^: coefficient of determination; RMSEC/CV/P: root mean square error of calibration/cross validation/prediction; Q^2^: predictive R^2^; lof: lack of fit; RPD: residual predictive deviation; LOO-CV: leave one out cross validation; 5CV: five-fold cross validation.

Univariate Linear Regression Analysis (ULRA)									
Y Variable	N	Intercept	slope	R^2^	RMSEC	Q^2 a^	RMSECV ^a^	RPD ^a^	RMSEP ^d^
Proteins	52	3.793	145.371	0.876	3.211	0.859	3.417	2.693	3.186
Lipids	50	−0.027	149.447	0.816	1.189	0.798	1.246	2.248	0.726
Carbohydrates	46	0.426	66.945	0.647	4.413	0.611	4.634	1.621	9.33
Carbohydrates/Proteins	46	0.027	0.375	0.844	0.428	0.833	0.442	2.476	0.684
**Orthogonal Partial Least Squares (OPLS)**									
**Y Variable**	**N**	**Components ^b^**	**R^2^X(cum)**	**R^2^Y(cum)**	**RMSEC**	**Q^2^(cum) ^c^**	**RMSECV ^c^**	**RPD ^c^**	**RMSEP ^d^**
(model)	46	2 + 2 + 1	0.984	0.861		0.837			
Proteins	46			0.916	2.944	0.898	3.073	2.994	1.48
Lipids	46			0.901	0.932	0.877	0.979	2.859	1.135
Carbohydrates	46			0.768	3.797	0.735	3.826	1.964	4.081
**Multivariate Curve Resolution Alternating Least Squares (MCR–ALS)**									
**Y Variable**	**N**	**lof PCA (%)**	**lof exp (%)**	**R^2^**	**RMSEC**	**RMSECV ^a^**			
(model)	52	0.798	5.854	0.997					
Proteins	52			0.851	3.521	3.699			
Lipids	50			0.768	1.335	1.392			
Carbohydrates	46			0.632	4.508	4.73			

^a^ LOO-CV. ^b^ Predictive + Orthogonal in X + Orthogonal in Y. ^c^ 5CV. ^d^ external validation.

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
