# Peer review of "Statistical Methods for Rapid Quantification of Proteins, Lipids, and Carbohydrates in Nordic Microalgal Species Using ATR–FTIR Spectroscopy"

_molecules, 2019, doi:10.3390/molecules24183237_

Round 1

Reviewer 1 Report

The manuscript describes a novel approach for quantification of major biochemical constituents of microbial biomass. The authors provide comprehensive study by infrared spectroscopy and chemometrics for chemical characterisation of seven green microalgae. The study offers reliable methodology for biomass analysis of relevance to the microbiology and biotechnology research, in particular monitoring and screening studies.

The manuscript is well write, concise, and measurements are sufficiently supported with facts and figures. Therefore, I recommend that the manuscript is accepted for publication after a minor revision.

The more detailed comments are as follows:

Line 82: avoid instead of avoiding

Line 132-136: Better explanation for discrepancy between spectral and reference (extraction) measurements is needed in order to justify exclusion of some data from the statistical analyses. According to Table A2, the extraction results are very reproducible. While the concentration of carbohydrates in C. vulgaris 13-1 at day 0 can be considered as an outlier, the amount of lipids and carbohydrates in C. astroideum RW10 at day 6 and 8 is within normal range of values. In addition to the justification for exclusion of data, I recommend that the corresponding FTIR spectra are added to the Appendix.

Figure 1: X-axis should be reversed (1800-800 cm-1). The standard convention in optical spectroscopy is to display spectra from lower to higher wavelengths (i.e. higher to lower wavenumbers).

Table A1: SP strain is missing from the table legends.

Table A1 and A2: For clarity, it would be better if the strain names were directly stated in the table, instead of non-intuitive codes.

Author Response

The manuscript is well written, concise, and measurements are sufficiently supported with facts and figures. Therefore, I recommend that the manuscript is accepted for publication after a minor revision.

We thank reviewer 1 for the positive evaluation of our manuscript.

Line 82: avoid instead of avoiding

We made the correction, as suggested.

Line 132-136: Better explanation for discrepancy between spectral and reference (extraction) measurements is needed in order to justify exclusion of some data from the statistical analyses. According to Table A2, the extraction results are very reproducible. While the concentration of carbohydrates in C. vulgaris 13-1 at day 0 can be considered as an outlier, the amount of lipids and carbohydrates in C. astroideum RW10 at day 6 and 8 is within normal range of values. In addition to the justification for exclusion of data, I recommend that the corresponding FTIR spectra are added to the Appendix.

Based on our observations, these data had a large negative effect on the regression analysis both in terms of data fitting (R2) and prediction power (Q2). This in part is due to the comparatively small size of the dataset used for building the models (n = 52). However, we assumed that the observed discrepancy between spectral and empirical data was coming from an experimental error in the conventional chemical extraction and quantification, and not from the spectral recording, as FTIR measurements for these samples were repeated and provided nearly identical results.

We improved the explanation for the detection of these outliers and their exclusion from the statistical analysis, as follows:

“Compound amounts differing between spectroscopic analysis and biochemical extraction were considered as outliers and excluded from the statistical analyses: the amount of lipids and carbohydrates in C. astroideum RW10 at the end of the N starvation period (day 6, 8) appeared overestimated after chemical extraction, likely due to a carryover of impurities after chemical extraction, while the concentration of carbohydrates in C. vulgaris 13-1 at day 0 of the N starvation period was probably underestimated using this method, likely due to an inefficient cell wall hydrolyzation. The exclusion of these outliers (Table A2) considerably improved the fitting of data and predictive power of the models (i.e. higher R2 and Q2) for lipids and carbohydrates with all the statistical methods tested (data not shown). FTIR spectra of the excluded samples are given in the Appendix (Figure A1). While spectra from the same culture collected in previous and consecutive sampling match, the empirical values deteriorate. This points to an experimental error in the empirical values rather than in the spectral recording.”

A new figure with the corresponding FTIR spectra has been added to Appendix 1 (Figure A1), following Reviewer’s 1 suggestion. Please, note that accordingly Figure A1 in the previous version of the manuscript has been now renamed as Figure A2 in the current version.

Figure 1: X-axis should be reversed (1800-800 cm-1). The standard convention in optical spectroscopy is to display spectra from lower to higher wavelengths (i.e. higher to lower wavenumbers).

The X-axis of Figure 1 has been now reversed, as requested by Reviewer 1.

Table A1: SP strain is missing from the table legends.

The name of the strain Scenedesmus obliquus has been now corrected to “So UTEX” in Table A1.

Table A1 and A2: For clarity, it would be better if the strain names were directly stated in the table, instead of non-intuitive codes.

For brevity, we decided to keep the codes for the algal strains, rather than using the extended names in the tables. However, for further clarity, we added acronyms to each strain ID, e.g. Cv for Chlorella vulgaris, and we described the code usage in the captions accordingly for easier understanding.

Reviewer 2 Report

The authors reported on ATR-FTIR spectroscopy as a novel technique to evaluate the chemical composition (proteins, lipids and carbohydrates) and its changes in complex biological systems. The obtained results were analysed by means of three statistical methods.

The manuscript was excellently written (in my opinion) and I detected only one minor error. The manuscript was written and the study planned and executed in a logical and systematic manner, with an in-depth discussion on the successes and shortcomings of the statistical methods used.

Suggested corrections:

Spelling error: p. 9, line 312, glass "beats" should read glass "beads"

Author Response

The authors reported on ATR-FTIR spectroscopy as a novel technique to evaluate the chemical composition (proteins, lipids and carbohydrates) and its changes in complex biological systems. The obtained results were analysed by means of three statistical methods.

The manuscript was excellently written (in my opinion) and I detected only one minor error. The manuscript was written and the study planned and executed in a logical and systematic manner, with an in-depth discussion on the successes and shortcomings of the statistical methods used.

The authors acknowledge the very positive evaluation of the manuscript from Reviewer 2.

Suggested corrections:

Spelling error: p. 9, line 312, glass "beats" should read glass "beads"

The spelling error has been now corrected, as suggested.

Reviewer 3 Report

The manuscript under review is a well written work. The results are of interest. The only significant concern that should be addressed is regarding the biochemical composition of the microalgae (table A2), where there are some species that give a total  amount of lipids, proteins and carbohydrates less than 60%. Why is this high deviation on the results and how do they effect the FTIR spectra and the relationship between FTIR and biochemical compositions afterall?

Author Response

The manuscript under review is a well written work. The results are of interest. The only significant concern that should be addressed is regarding the biochemical composition of the microalgae (table A2), where there are some species that give a total amount of lipids, proteins and carbohydrates less than 60%. Why is this high deviation on the results and how do they effect the FTIR spectra and the relationship between FTIR and biochemical compositions afterall?

The values measured in this work for lipids, proteins and carbohydrates are in line with those typically found in microalgae and the sum of these compounds can give values well below 60 % in some species (see Molino et al. 2018). In fact, ash and fibres can represent a considerable part of the algal biomass, accounting to more than 50 % of the dry weight. However, as discussed in the manuscript, underestimation in the concentration of these compounds is mostly related to the limitations of the conventional extraction and analytical methods not to the spectral profiles. Some algal species showed to have strong cell walls, which affected the extraction efficiency. Since these values are used to fit (calibrate) the spectral profiles, an offset in them carries over to the models. The purpose of this work was to demonstrate the advantage of using FTIR spectroscopy, as this alternative method relies on the direct measurement of the biomass sample as a whole, with no need for pre-treatment and extraction steps.

Reference:

Molino, Antonio et al. 2018. “Microalgae Characterization for Consolidated and New Application in Human Food, Animal Feed and Nutraceuticals.” International Journal of Environmental Research and Public Health 15(11): 1–21.